# Nutritional Parameters, Biomass Production, and Antioxidant Activity of *Festuca arundinacea Schreb.* Conditioned with Selenium Nanoparticles

**DOI:** 10.3390/plants11172326

**Published:** 2022-09-05

**Authors:** Uriel González-Lemus, Gabriela Medina-Pérez, José J. Espino-García, Fabián Fernández-Luqueño, Rafael Campos-Montiel, Isaac Almaraz-Buendía, Abigail Reyes-Munguía, Thania Urrutia-Hernández

**Affiliations:** 1Instituto de Ciencias Agropecuarias, Universidad Autónoma del Estado de Hidalgo, Av. Rancho Universitario s/n Km. 1, Tulancingo C.P. 43600, Hidalgo, Mexico; 2Sustainability of Natural Resources and Energy Program, Cinvestav-Saltillo, Ramos Arizpe C.P. 25900, Coahuila de Zaragoza, Mexico; 3Unidad Académica Multidisciplinaria Zona Huasteca, Universidad Autónoma de San Luis Potosí, Romualdo del Campo No. 501, Fracc. Rafael Curiel, Ciudad Valles C.P. 79060, San Luís Potosi, Mexico

**Keywords:** grass, flavonoids, foliar, forage, phenols, zacate fescua

## Abstract

*Festuca arundinacea Schreb.* is a widely used type of forage due to its great ecological breadth and adaptability. An agricultural intervention that improves the selenium content in cultivated plants has been defined as bio-fortification, a complementary strategy to improve human and non-human animals’ nutrition. The advancement of science has led to an increased number of studies based on nanotechnologies, such as the development of nanoparticles (NPs) and their application in crop plants. Studies show that NPs have different physicochemical properties compared to bulk materials. The objectives of this study were (1) to determine the behavior of *F. arundinacea*
*Schreb.* plants cultivated with Se nanoparticles, (2) to identify the specific behavior of the agronomic and productive variables of the *F. arundinacea Schreb.* plants, and (3) to quantify the production and quality of the forage produced from the plant (the bioactive compounds’ concentrations, antioxidant activity, and the concentration of selenium). Three different treatments of SeNPs were established (0, 1.5, 3.0, and 4.5 mg/mL). The effects of a foliar fertilization with SeNPs on the morphological parameters such as the root size, plant height, and biomass production were recorded, as well as the effects on the physicochemical parameters such as the crude protein (CP), lipids (L), crude fiber (CF), neutral detergent fiber (NDF), acid detergent fiber (ADF), carbohydrates (CH), the content of total phenols, total flavonoids, tannins, quantification of selenium and antioxidant activity 2,2′-Azino-bis(3-ethylbenzothiazoline-6-sulfonic acid) (ABTS), and 2,2-diphenyl-1-picrylhydrazyl (DPPH). Significant differences (*p* < 0.05) were found between treatments in all the response variables. The best results were obtained with foliar application treatments with 3.0 and 4.5 mg/mL with respect to the root size (12.79 and 15.59 cm) and plant height (26.18 and 29.34 cm). The *F. arundinacea Schreb.* plants fertilized with 4.5 mg/L had selenium contents of 0.3215, 0.3191, and 0.3218 mg/Kg MS; total phenols of 249.56, 280.02, and 274 mg EAG/100 g DM; and total flavonoids of 63.56, 64.96, and 61.16 mg QE/100 g DM. The foliar biofortified treatment with a concentration of 4.5 mg/mL Se NPs had the highest antioxidant capacities (284.26, 278.35, and 289.96 mg/AAE/100 g).

## 1. Introduction

*Festuca arundinacea* (*F. elatior L. ssp. arundinacea* (*Schreber.*) Hackel) is a temperate climate forage highly valued for its hardiness, phenotypic plasticity, and quality in extensive livestock production systems, especially those involving ruminants [1,2]. *F. arundinacea** Schreb.* is widely used as forage due to its great ecological breadth and adaptability; it is also helpful in soil conservation due to its extensive and penetrating root system. This plant is mainly used in the winter, during which its most significant value arises [3,4]. Selenium (Se) is an essential micronutrient for animals; However, part of the area used for agriculture contains low levels of this trace element. Additionally, Se can have beneficial effects on the growth and performance of plants since it can increase their tolerance to different types of stress and generate the synthesis of phytochemicals with antioxidant properties [5,6]. Thus, Se can be present, but in unavailable chemical forms for organisms; as a result, there is a deficiency in the Se content in plants, animals, and other organisms in the food chain [7,8]. Selenium is metabolized in plants by the assimilation of sulfur, and its distribution and accumulation depend on the chemical species [9,10]. Plants can absorb Se as selenate (Se^6+^), selenite (Se^4+^), and organic selenite. Both selenate (Se^6+^) and organic Se are metabolically active, while selenite (Se^4+^) can remain a passive component. Organic forms of Se are more available to plants than inorganic forms [11]. In Se accumulator plants, Se is metabolized in plants by the assimilation of sulfur and its distribution and accumulation depend on the chemical species. Selenium is incorporated through the alternative sulfate route because it is chemically similar and acts analogously in many biochemical reactions. Se is metabolized in plants by sulfur assimilation, and its distribution and accumulation depend on the chemical species. In Se accumulator plants, it is incorporated via the alternate route of sulfate because it is chemically similar and acts analogously in many biochemical reactions [12]. Almost all crops, including tomatoes, are of the so-called non-Se accumulator type; that is, they are plants for which more than 25 µg g^−1^ of Se (dry weight) in the roots and leaves results in toxicity [13,14]. Although incorporating Se in an edaphic form is the common alternative, several factors influence the accumulation of Se in plants (e.g., the concentration and type of Se, pH, redox potential, and the concentration of other ions in the soil). In addition, plant factors such as membrane transport activity or translocation mechanisms also influence the absorption of Se [15]. Some agricultural experiments that improve Se content in cultivated plants have been defined as bio-fortification, which is a complementary strategy to improve human and other animals’ nutrition [16]. Biofortification has been carried out in agricultural products (wheat, sorghum, peas, rice, and vegetables) and in some forage varieties such as alfalfa that lack Se due to the low concentrations of this trace element in soil [17]. There are several bio-fortification methods used for plants, but the foliar application method has proven to be more effective than soil fertilization [18]. In recent years, the benefits that Se provides for the health of farm animals—improving reproductive performance and immune and homeostatic function—have been demonstrated [19]. Worldwide, several studies relate the absence of the consumption of this element with the decrease in development, growth, and with reproductive and metabolic problems, especially in ruminants [20,21]. The advancement of science has led to increased studies focused on nanotechnologies, such as the development of nanoparticles (NPs) and their application in crop plants. Studies show that NPs have different physicochemical properties compared to bulk materials.

Black and gray elemental Se are insoluble. Red selenium nanoparticles (SeNPs) exert scavenging effects on various free radicals in vitro; however, Se is insoluble and has little bioactivity. SeNPs are used as antibacterial, antioxidant, and anticancer agents in medicine. In trials with rats and crucian carp, SeNPs were more successful than SeMet at increasing the activities of glutathione peroxidase (GSH-Px), thioredoxin reductase (TrxR), and glutathione Sansferase (GST), while SeNPs exhibited a lower cytotoxicity in mice than selenite. 

## 2. Results and Discussion

### 2.1. Biomass Production and Phenological Characteristics of F. arundinacea Schreb. Plants

Table 1 and Figure 1 shows the results of the treatments’ biomass production, root length, and the length of the leaves. Significant differences exist (*p* < 0.05) between the treatments and the control. The increase in the length of roots, the length of the leaves, and the biomass production of the Se-treated plants could be attributed to the application of selenium nanoparticles. Selenium is a stimulator of photosynthetic pigments and organogenesis. In [6,22,23,24], the authors found that Se increases the growth of lettuce and green tea crops. Ref. [22] applied selenium in tomatoes and found no differences between the different treatments’ fresh and dry weights of roots. In ryegrass, selenium has been shown to promote the growth of roots [23]. Selenium hyperaccumulation may offer better growth, perhaps due to the better resistance to oxidative stress. The authors of [6,25] documented the effects of nanoparticles on bitter melon (*Momordica charantina* L.) seeds and confirmed the plant’s nanoparticle (NP) capacity, translocation, and accumulation. In addition, they found an increase of 54% in biomass production, 24% in water content, and an increase of 20% in fruit length. Ref. [24] observed that the foliar application of different amounts of NPs of Se affected the morphological characteristics such as the shoot length, root size, and fresh weight of a bean plant. In ref. [26], the authors applied sodium selenate and Se nanoparticles at 3 mg/L to tomato plants. In all the treatments, the fresh biomass in the shoot increased significantly by 23% and 35% when applied with both types of Se. The same authors documented significant differences between the treatments (regarding the plant’s vegetative, reproductive, and flowering stages) when selenium was applied (in bulk or as a nanoparticulate).

### 2.2. Nutritional parameters of F. arundinacea Schreb. Plants

In all the determinations, significant differences (*p* < 0.05) were observed between treatment four and the control. No significant differences were observed between the different harvests established. The results obtained in this study are similar to those reported in [27], where Se was used as a fertilizer, and its application increased the protein content of the roots and leaves of alfalfa (*Medicago sativa* L.). Similarly, Ref. [25] found that the foliar spraying of selenate increased the protein and total nitrogen content of radish roots (*Raphanus sativus* L.). Ref. [24] observed that the foliar application of different amounts of NPs of Se affected the morphological characteristics such as the shoot length, root size, and fresh weight of the bean plant.

Table 2 shows proximal analyses composition of *Festuca arundinacea Schreb*. plants. The increase in the CP content—as the plants’ response to the foliar biofortification of the Se NPs—manifests because Se promotes the increase in the synthesis of sulfur amino acids (Cys and Met) and the selenium-amino acids Se-Cys and Se-met, which are incorporated into proteins. However, there is also evidence that other non-protein amino acids such as γ-glutamyl methyl selenium cysteine (γ-gluMetSeCys), methyl-SeCys, and methyl-Semet are synthesized. The authors of [28,29] obtained results wherein the foliar supplementation of sodium selenate in wheat plants (*Triticum aestivum L.* variety BRS 264) increased the total nitrogen content (up to 20%) compared to the control.

Garcia et al. [28] reported that fertilization with Se promotes the increase in fatty acids (oleic, linoleic, and linolenic), in turn increasing the concentration of lipids; however, the metabolic pathway is unclear. Lindom et al. [30] found that the foliar application of sodium selenite and selenate in four rice genotypes (Ariete, Albatros, OP1105, and OP1109) increased the total lipid content, as well as the content of oleic acid (C18:1), linoleic (C18:2), and palmitic acid (C16:0). The decrease in CF can be attributed to the fact that Se-treated plants have been shown to increase the concentration of soluble sugars. This could consequently decrease the presence of structural carbohydrates because Se regulates carbohydrates’ metabolism through the altered redox potential that can stimulate nodulation [31], and in turn generate a higher CO_2_ fixation due to the higher stomatal conductance. Kaur et al. [31] reported on the synthesis of compounds such as starch, total soluble sugars, and reducing sugars caused by the possible increase in the activity of enzymes involved in carbohydrate metabolism. Lara et al. [29] found that the carbohydrate content in wheat plants increased with Se biofortification, showing this trend in different crops.

### 2.3. Bioactive Compounds

#### Total Content of Phenols, Flavonoids, and Tannins

The extracts of the different treatments in the different crops showed significant differences (*p* < 0.05) in the content of total phenols, flavonoids, total tannins, and selenium (Table 3). The foliar application of NPs of Se resulted in an increased (*p* < 0.05) total phenol content compared to the value of the control treatment. The extract with the highest phenol content was the biofortified one with 4.5 mg of Se NPs, presenting this trend in each extract from the different harvests between treatments (249.56, 280.02, and 274.63 mg GAE/100 g).

According to Garcia et al. [28], the higher total phenols in the treatments fortified with Se NPs can be attributed to selenium. It could modify the phenylpropanoid pathway by increasing the enzymatic activity of phenylalanine ammonium lyase (PAL). In several investigations, a correlation was found between Se content and the increase in phenolic compounds [28]. For example, ref. [28] conducted a study on olive oil; found significantly higher levels of phenolic compounds due to the observed inducing effects on PAL activity due to Se biofortification. Similarly, Schiavon et al. [28] supplied selenate in tomatoes (*Solanum lycopersicon* L.) through the roots, and phenolic compounds were synthesized in the leaves. The results obtained are similar to those described by [32], where watercress (*Lepidium sativum* L.) seeds were bio-fortified with Se NPs, and it was found that concentrations of 50 mg significantly increased the content of the total phenols (4896 mg GAE/100 mg). On the other hand, Ref. [33] carried out a study in which the authors applied different concentrations of Se NPs in three crops of Peanuts (*Arachis hypogaea* L.) (NC, Gregory and Giza 6). The results showed that the total content of phenols increased (1.4 mg) in the NC and Giza 6 treatments compared to the control treatment (1.0 mg).

On the other hand, the increase in the number of flavonoids and tannins is related to the concentration of Se. As a result, the extracts of the grass sprinkled with 4.5 mg of NPs presented the highest content of flavonoids (63.56, 64.96, and 61.16 mg QE/100 g) and a higher tannin content (0.32, 0.31, and 0.32 mg EC/100 g) in comparison to other treatments (*p* < 0.05) and the control. Flavonoids are low molecular weight compounds found in plants that help stabilize the free radicals that generate stress [34].

The high flavonoid content of the Se-treated plants could be attributed to the concentrations of NPs of Se applied via foliar spraying on *Festuca arundinacea*
*Schreb.* plants). According to Schiavon et al. [34], the levels of naringenin, chalcone, and kaempferol, among the other flavonoids present in Tomatoes (*Solanum lycopersicon* L.), were due to a positive response from the application of Se. The results obtained in this investigation are similar to those reported by [34], where the fortification of Se NPs (5 mg/L) in celery (*Apium graveolens* L.) increased the total content of flavonoids by 1.5 times compared to the control. In the study by [34], the flavonoid content doubled (69.7 mg/100 g) in *Brassica juncea* L. due to foliar selenium biofortification compared to the control (31.2 mg/100 g).

On the other hand, significant differences (*p* < 0.05) were found between the different forage extracts foliar-fortified with NPs of Se and the control in each treatment concerning the total tannins content. The extracts of the treatments sprayed via the foliar route with 4.5 mg NPs of Se presented the highest tannin content (151.39, 172.41, and 175.77 mg/100 g) compared to the control treatment (101.16, 109.16, and 119.11 mg/100 g). The results are similar to those reported by [35]. The authors soaked buckwheat seeds (*Fagopyrum esculentum* Moench) in sodium selenate (5, 10, and 20 mg/L) and sodium selenite (10 and 20 mg/L); as a result, the tannin content in the leaves and stem increased in the Se-treated plants. It has been shown that NPs of Se have a greater enrichment capacity than other forms of selenium supplementation [34].

### 2.4. Selenium Content

The selenium content in the biofortified forage treatments increased significantly via foliar application (*p* > 0.05) between treatments. There were no significant differences in the concentration of Se (*p* < 0.05) in each treatment compared to the different crops. The accumulation of Se in the treatment with 4.5 mg of Se NPs was approximately four times more (0.3215, 0.3191, and 0.3218 mg/kg DM) compared to the control, showing this trend in the three pasture crops. The high content of Se in forage grasses is significant since ruminants require 0.1 to 0.3 mg kg^−1^ DM of Se; the requirements depend on the weight, diet, and physiological state of the animal. Selenium is part of selenium-cysteine, an oxide-reducing protein of amino acids. The low thyroid hormone level affects the metabolism, and the immune response is altered, causing muscular pathologies and reproductive disorders [34]. Se uptake and accumulation in forage (*Festuca arundinacea Schreb*.) show a dose-response relationship with Se NPs when applied via foliar. One study [34] reported an experiment with spinach (*Spinacia oleracea* L.). The authors found an increase in Se content due to foliar Se enrichment. Ref. [34] enriched rice with selenite and sodium selenate by the foliar route. The effects of the fertilization were higher when selenite (0.471 µg g^−1^) and selenate (0.640 µg g^−1^) were applied compared to the control (0.071 µg g^−1^).

### 2.5. Antioxidant Activity by Inhibiting the Radical DPPH and ABTS

Significant differences were found in the antioxidant activity by inhibiting the radical ABTS (*p* < 0.05) (Table 4). All the extracts of grass enriched with NPs of Se showed an increase in antioxidant activity in the ABTS radical inhibition test. The fortified foliar treatment with a concentration of 4.5 mg/L NPs presented the highest antioxidant capacity (284.26, 278.35, and 289.96 mg AAE/100 g) with significant differences from the other treatments (*p* < 0.05).

The results of the antioxidant activity determined by the measurement of the DPPH radical are shown in Table 3. These results are similar to those for ABTS radical inhibition. It is essential to mention that all the treatments fortified with NPs of Se presented a high (*p* < 0.05) antioxidant activity when compared to the control. The increase in antioxidant activity found in the biofortified treatments with NPs of Se could be attributed to a better synthesis of the compounds (phenols and flavonoids) that belong to the non-enzymatic antioxidant system. Flavonoids are secondary metabolites with regulatory functions in plant development, UV protection, and defense mechanisms that act as antioxidants in plants [34]. Phenolic compounds act as high-level antioxidants because they scavenge free radicals and active oxygen species, such as singlet oxygen, superoxide free radicals, and hydroxyl radicals [34]. This radical scavenging activity is attributed to substituting hydroxyl groups in the aromatic ring systems of phenolic compounds due to their ability to donate hydrogen [34]. On the other hand, Ref. [36] showed that NPs of Se could scavenge ABTS radicals. Ref. [37] obtained results that show that the extract of alfalfa (*Medicago sativa* L.), a plant used as fodder for livestock, contains polyphenolic compounds, which present antioxidant activity, and this was demonstrated by the ABTS and DPPH free radical-scavenging tests. Alternatively, ref. [38] found that the extracts of five geophytes (*C. capitatus* and *C. conglomeratus*, *E. farctus*, *L. indicas*, and *p. turgidum*) showed a significant increase in the inhibitory activity of 2,2-diphenyl-1-picrylhydrazyl (DPPH). Those extracts showed the inhibitory effect of the radical due to a high content in the composition of several secondary metabolites (total phenolics, tannins, total flavonoids, alkaloids, and saponins). Ref. [39] studied different cultivars of oats (*Avena sativa*) and found that one of the treatments (OS-6 cv) had the highest inhibition towards the radicals 2, 2-diphenyl-1′picrylhydrazyl (DPPH) and 2,2′-azinobis (3-ethylbenzothiazoline-6-sulfonic acid) (ABTS).

## 3. Materials and Methods

Common “zacate fescua” (*Festuca arundinacea Schreb*.) was grown in a greenhouse and was fertilized via the foliar application with Se nanoparticles. The response of the plants grown for 90 days was measured.

### 3.1. Biological Material

Common “zacate fescua” (*Festuca arundinacea*
*Schreb.*) seeds were purchased from “El Instituto Nacional de Investigaciones Forestales, Agrícolas y Pecuarias” (INIFAP-Celaya, Mexico). The seeds were stored in the dark at 4 °C until use.

### 3.2. Nanomaterials

Nanoparticles of selenium were purchased from Materiales Nanoestructurados S.A de C.V. (San Luis Potosí, México). Physicochemical characteristics of selenium nanoparticles are listed in Table 5.

### 3.3. Experimental Design and Greenhouse Experiment

The experiment’s soil was obtained from the Institute of Agricultural Sciences of the Autonomous University of Hidalgo State (Hidalgo State, Mexico) from the rotated crop areas. The geographic coordinates were latitude north 20°04’53’’ and latitude west 98°22’07’’ of the meridian of Greenwich. According to FAO/UNESCO soil classification system, the soil was a haplic *phaeozem* with pH 7.54 and electrolytic conductivity of 5.3 dS m-1, a water holding capacity (WHC) of 625.01 g kg^−1^, an organic carbon content of 3.6 g C kg^−1^ soil, and a total inorganic N content of 0.21 g N kg^−1^ soil.

The experiment was performed in a greenhouse at the Institute of Agricultural Sciences of the Autonomous University of Hidalgo State. First, a completely randomized block design was used. Common grass seeds (*Festuca arundinacea*
*Schreb.*) with a certified quality by “El Instituto Nacional de Investigaciones Forestales, Agrícolas y Pecuarias” (INIFAP) were planted. Subsequently, with a manual rounder, sowing was carried out with 40 kg/ha density in triplicate for each treatment. The temperature during the experiment was 25–28 °C during the day and 16–19 °C at night from May to September 2020.

### 3.4. Foliar Application of Selenium Nanoparticles

Biofortification with selenium in *Festuca arundinacea Schreb.* was performed through a foliar application with NPs of Se in different concentrations (1.5, 3.0, and 4.5 mg/L). Distilled water was applied to the control. The foliar application of Se NPs was carried out 21 days after seed germination, during the vegetative phase of the crop, following the methodology proposed by [40] with some modifications. Three times of harvest of the grass were established during the 90 days. The Se NPs with a size <50 nm were acquired from ID-nano Investigation y Desarrollo de Nanomateriales, SA de CV.

### 3.5. Measurement of the Growth of Festuca arundinacea Schreb. Plants

The plants of *Festuca arundinacea Schreb*. (8 biological repetitions) from the different treatments and the control were collected to measure the grass’s height and the root’s length with a vernier. Afterward, the harvest was carried out in triplicate for each treatment (1.5, 3.0, and 4.5 mg/L and the control). The fresh biomass production of each treatment was weighed.

### 3.6. Proximal Chemical Analysis of Festuca arundiceae Schreb. Plants

The proximal chemical analysis in the different pasture treatments (*Festuca arundiceae*) was performed in triplicate following the Association of Official Analytic Chemists (AOAC) [15]. First, the dry matter content (DM) was calculated by drying 5 g of grass in an oven (Craft USA) at 100 °C for 8 h (Official Method 925.09). The mineral content (MI) was determined by incinerating the samples in a muffle (FELISA model Fe 340) at 550 °C for 8 h (Official method 923.03). The ether extract (EE) was determined by the Soxhlet method (Official Method 923.05). The content of crude protein (CP) was measured by the Kjeldahl method (Method Official 981.10) (AOAC, 2000). Finally, the determination of neutral detergent fiber (NDF), acid detergent fiber (ADF), and lignin (ADL) was calculated following the methodology proposed by [41].

### 3.7. Extraction of Bioactive Compounds

Bioactive compounds were extracted according to the methodology of [41] with some modifications. First, 100 g of grass was placed in a drying oven with airflow (Craft HFA-1400 DP, USA) at 45 °C for 72 h. Subsequently, the samples were pulverized in a turbine mill to a particle size of 2 mm. A total of 40 mL of an ethanol/water solution (1:1) was added to 10 g of dry grass and left to macerate for 24 h with mechanical stirring (75 rpm) in the dark, and centrifuged at 18,510 *g* for 10 min at 4 °C. The supernatant was stored in 10 mL tubes at −70 °C without light until the samples were analyzed.

### 3.8. Determination of Total Phenols

The total phenol content was determined using the Folin–Ciocalteau method according to [42] with some modifications. In a test tube, 0.5 mL of grass extract and 2.5 mL of 10% Folin–Ciocalteu reagent(v/v) were added and incubated in the dark for 8 min, and 2.0 mL of Na_2_CO_3_ 7.5% (p/v) was added. After the mixture was vortexed (Vortex WM-10), it was allowed to react for 60 min in the dark at 22 °C. Finally, the absorbance at 765 nm was measured with a spectrophotometer (Jenway 6715, Staffordshire, ST15 OSA, UK) using distilled water as a blank. A calibration curve was developed using gallic acid as a standard (0–100 mg L^−1^), and the content of total phenols was expressed as equivalent mg of gallic acid per 100 g of dry sample (mg GAE/100 g).

### 3.9. Determination of Total Flavonoids

The determination of flavonoids was carried out according to [43] with some modifications. Grass extract of 0.5 mL was used with 0.5 mL of a 10% (*p*/*v*) aluminum trichloride (AlCl_3_) solution and 0.5 mL of 0.1 mM sodium nitrate. Subsequently, the mixture was vortexed and incubated for 60 min at 22 °C. Then, 250 µL of NaOH (1 M) was added to stop the reaction. After incubation, the absorbance of the mixture was measured at a wavelength of 415 nm using a UV-VIS spectrophotometer (Jenway 6715, Staffordshire, ST15 OSA, UK). A calibration curve was developed using quercetin as a standard (0–100 mg L^−1^) and methanol as blank; the results were expressed in mg equivalents of quercetin per 100 g dry sample (mg QE/100 g).

### 3.10. Determination of Tannins in Grass

The tannin content was measured according to [44] with some modifications. A 0.1 M FeCl_3_ solution was made using 0.1 M HCl as solvent. Then, 600 µL of FeCl_3_ was added to the test tube, 200 µL of the grass extract was left for 5 min in the dark, and the mixture was allowed to react for 10 min at 22 °C. The samples were read with a UV-VIS spectrophotometer (Jenway 6715, Staffordshire, ST15 OSA, UK) at 720 nm. A treatment was prepared with the same conditions by replacing the grass extract with an ethanol/water solution (1:1), which was analyzed. The result was subtracted from the readings of the different treatments. A catechin standard curve was developed, and the results were expressed as mg equivalents of catechin per 100 g of dry sample (mg EC/100 g).

### 3.11. Antioxidant Activity of Forage

#### 3.11.1. Determination of Antioxidant Activity by Inhibiting the Radical DPPH

According to [45] methodology (Brand-Williams et al., 1995), 1,1-diphenyl-2-picrylhydrazyl radical (DPPH) was used with some modifications. First, it was prepared with an 80% methanol solution at a concentration of 6.1 × 10^−5^ mol/L. Subsequently, mechanical stirring was performed at 30 rpm for 120 min in total darkness. For the test, a mixture was made using 0.5 mL of the grass extract and 2.5 mL of DPPH solution, which was allowed to react for 60 min in the dark, and it was read at 515 nm in a spectrophotometer (Jenway 6715, Staffordshire, ST15 OSA, UK). A blank with 80% methanol and water was used, and the mixture was read at 515 nm. A calibration curve was developed using ascorbic acid as a standard. The results were expressed as mg equivalents of ascorbic acid per 100 g dry forage (AAE/100 g DM).

#### 3.11.2. Determination of Antioxidant Activity by Inhibition of the Radical ABTS

The 2,20-azino-bis (3-ethylbenzothiazoline-6-sulfonic acid) radical (A1888 Sigma-Aldrich^®^, Saint Louis, MO, USA) was used to measure antioxidant activity. A total of 10 mL of the 7 mM ABTS solution with 10 mL of 2.45 mM potassium persulfate (K_2_S_2_O_8_) (Sigma-Aldrich^®^ 216224, Saint Louis, MO, USA) were reacted together. The mixture was stirred in the dark for 16 h. The ABTS radical was adjusted with a 20% ethanol solution (100983 Merck^®^, Kenilworth, NJ, USA) to an absorbance of 0.7 ± 0.1 and a wavelength of 734 nm; subsequently, 200 µL of the extract was added to 2 mL of adjusted ABTS. The reaction occurred for 6 min [46]. A calibration curve was developed using ascorbic acid as a standard. The results were expressed in mg equivalents of ascorbic acid for 100 g dry forage (AAE/100 g DM).

### 3.12. Measurement of Selenium Content in Grass

The analysis of Se content in the grass (*Festuca arundinacea*
*Schreb.*) was carried out using the acid digestion procedure, pre-reduction of Se, and detection by hydride generation atomic absorption spectrometry (HG-AAS), following the methodology of [47] with some modifications. A total of 2 g of dry, ground, and sieved forage (<2 mm) was placed in a digestion tube, then 20 mL of concentrated HNO_3_ was added, and the mixture was heated at 175 °C for 60 min. The temperature of the heated sample decreased to 150 °C and was incubated for 90 min. A total of 5 mL of a mixture of H_2_SO_4_ and HNO_3_ (2:1) was added to the tube, heated to 175 °C for 60 min, cooled to 20 °C, and then 2 mL of H_2_O_2_ was added dropwise and again heated to 140 °C for 10 min. The digestion result (whitish or slightly yellow) was diluted to 25 mL with double-distilled water. Subsequently, a pre-reduction of SeVI to SeIV was carried out, 5 mL of 6 M HCl was added to 5 mL of the previous mixture, and it was heated at 60 °C for 120 min. In all cases, the standard addition method was used. The detection of selenium in the samples was carried out using an atomic absorption spectrophotometer (Perkin-Elmer AAnalyst 400). The equipment had an air-acetylene flame atomizer, a Se hollow cathode lamp with a spectral slit width of 2.0 nm, and an absorption wavelength of 196 nm with a hydride generator (Perkin Elmer MHS-15) consisting of a peristaltic pump and injection valve. The acetylene gas flow was 2.5 L min^−1^ and the airflow 11.5 L min^−1^. The results were expressed as mg of Se per kg of dry matter.

### 3.13. Statistical Analysis

An analysis of variance was performed based on a completely randomized design. When significant differences (*p* < 0.05) were observed between the treatments, the Tukey method using the SigmaPlot 12.0^®^ software compared the means.

## 4. Conclusions

A selenium fertilization through nanoparticles had beneficial effects on forage grass *Festuca arundinacea*
*Schreb.* and its yield. The application of NPs of Se increases the content of crude protein and ether extract. Additionally, the content of phenols, flavonoids, tannins, and selenium increased significantly through the fertilization with NPs of Se. The antioxidant activity of the grass is higher due to the physiological effects produced in the grass by the foliar fertilization with NPs of Se. In addition, NPs of Se could be an excellent alternative to improve the production of higher quality forage crops.

Our results show that selenium affects the selenium content in the biomass and the bioactive compounds within the first two weeks after application. Therefore, selenium nanoparticles can be considered as an alternative form of selenium for plant nutrition.

## Figures and Tables

**Figure 1 plants-11-02326-f001:**
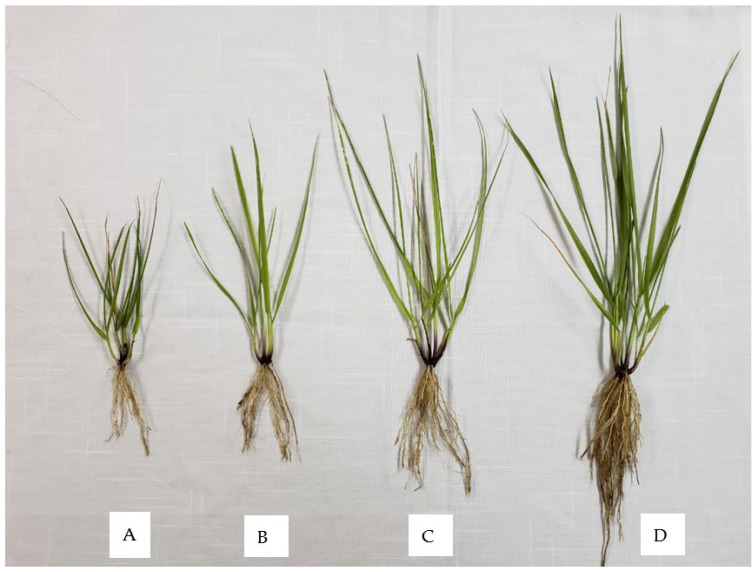
Effects of different concentrations of selenium nanoparticles (SeNPs) applied by foliar route on the characteristics of forage grasses (*Festuca arundinacea Schreb*.): (**A**) Control 0; (**B**) 1.5 ppm; (**C**) 3.0 ppm; (**D**) 4.5 ppm.

**Table 1 plants-11-02326-t001:** Effect of foliar application of Se NPs on the agronomic and productive variables of the *F. arundinacea*
*Schreb.* plants.

Treatment	
NPsSe (mg/L)	Biomass (kg)
0	0.196 ± 0.006 ^a^
2	0.213 ± 0.005 ^b^
3	0.238 ± 0.006 ^c^
4	0.275 ± 0.007 ^d^
Root (m)
0	0.069 ± 0.004 ^a^
2	0.074 ± 0.003 ^b^
3	0.116 ± 0.004 ^c^
4	0.159 ± 0.005 ^d^
Length (m)
0	0.145 ± 0.005 ^a^
2	0.198 ± 0.001 ^b^
3	0.245 ± 0.002 ^c^
4	0.332 ± 0.001 ^d^

The different lowercase letters ^(a–d)^ represents a significant difference between the treatments (*p* < 0.05).

**Table 2 plants-11-02326-t002:** Nutritional parameters of *Festuca arundinacea*
*Schreb.* bio-fortified via foliar with different concentrations of Selenium NPs.

Treatment NPsSe (mg/L)	CP (g/kg DM)	L (g/kg DM)	CF (g/kg DM)	NDF (g/kg DM)	ADF (g/kg DM)	CH (g/kg DM)
0	126.1 ± 0.95 ^c^	35 ± 0.17 ^a^	324.2 ± 0.26 ^c^	177.8 ± 0.14 ^c^	88.8 ± 0.28 ^c^	327.4 ± 0.20 ^c^
2	128.1 ± 0.63 ^c^	36 ± 0.20 ^a^	318.5 ± 0.53 ^b^	176.5 ± 0.05 ^c^	84.5 ± 0.09 ^b^	328.8 ± 0.10 ^b^
3	134.0 ± 0.16 ^b^	39 ± 0.20 ^b^	312.2 ± 0.48 ^b^	166.6 ± 0.16 ^b^	81.9 ± 0.08 ^b^	344.3 ± 0.05 ^a^
4	142.0 ± 0.36 ^a^	43 ± 0.30 ^c^	290.7 ± 0.25 ^a^	149.7 ± 0.31 ^a^	72.8 ± 0.37 ^a^	353.5 ± 0.53 ^a^

The lowercase letters ^a, b, c^ represent a significant difference between the treatments (*p* < 0.05 CP—crude protein, L—Lipids, CF—crude fiber, NDF—neutral detergent fiber, ADF—acid detergent fiber, and CH—carbohydrates.

**Table 3 plants-11-02326-t003:** Content of total phenols, flavonoids, total tannins, and selenium in the different treatments of foliar-biofortified grass (*Festuca arundinacea*
*Schreb.*) with different concentrations NPs of Se during the different times of harvests.

SeNPs (mg/L)	Total Phenols mg GAE/100 g DM
0	186.71 ± 7.51 ^a^
1.5	193.60 ± 5.06 ^a^
3.0	227.98 ± 8.63 ^b^
4.5	249.56 ± 7.89 ^c^
SeNPs (mg/L)	Total flavonoids mg QE/100 g DM
0	52.24 ± 1.82 ^a^
1.5	58.88 ± 1.23 ^b^
3.0	57.65 ± 1.63 ^b^
4.5	63.56 ± 1.28 ^c^
SeNPs (mg/L)	Total tannins mg EC/100 g DM
0	101.16 ± 2.38 ^a^
1.5	119.68 ± 4.12 ^b^
3.0	146.47 ± 3.18 ^c^
4.5	151.39 ± 4.68 ^d^
SeNPs (mg/L)	Selenium mg/kg DM
0	0.0715 ± 0.0086 ^a^
1.5	0.1197 ± 0.0082 ^b^
3.0	0.2578 ± 0.0092 ^c^
4.5	0.3215 ± 0.0132 ^d^

The different lowercase letters ^(a–d)^ in the columns represent a significant difference between the treatments. DM—Dry matter.

**Table 4 plants-11-02326-t004:** Displays the antioxidant activities by inhibiting the radical ABTS and DPPH of the grass extracts (*Festuca arundinacea Schreb*.) from treatments 0, 1.5, 3.0, and 4.5 mg/L of the SeNPs.

SeNPS mg/L	ABTS
0	252.86 ± 2.74 ^d^
1.5	274.51 ± 2.27 ^c^
3.0	293.01 ± 2.17 ^b^
4.5	312.78 ± 3.47 ^a^
	DPPH
0	217.53 ± 2.04 ^d^
1.5	225.04 ± 1.56 ^c^
3.0	257.24 ± 2.31 ^b^
4.5	284.26 ± 1.81 ^a^

The different capital letters in the same column indicate significant differences (*p* < 0.05) between each treatment on different analysis days. DPPH values are expressed in mg AAE/100 g, and ABTS are expressed in mg AAE/100 g.

**Table 5 plants-11-02326-t005:** Physicochemical characteristics of selenium nanoparticles used for foliar fertilization of the grass (*Festuca arundinacea*
*Schreb.*) in a 90-day greenhouse experiment.

Attribute	SeNPs
Chemical formula	Se
Color	Gray
Density (g/cm^−3^)	4.81
Molecular weight	78.96
Melting point	960.8 °C
Boiling point	222.12 °C
Magnetic properties	Weakly ferromagnetic
Particle size Morphology	Less than 100 nm Spherical 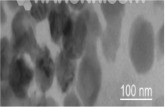

## Data Availability

Not applicable.

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
