# Peer review of "Nutritional Parameters, Biomass Production, and Antioxidant Activity of Festuca arundinacea Schreb. Conditioned with Selenium Nanoparticles"

_plants, 2022, doi:10.3390/plants11172326_

Round 1

Reviewer 1 Report

The reviewed manuscript contains the results of an interestingly designed experiment to evaluate the effects of foliar application of selenium nanoparticles (NPsSe) in forage (Festuca arundinacea) on morphological, physicochemical, nutritional and biomass traits in three different crops. Both the introduction to the research problem and the project assumptions were presented in a clear and essential manner, sufficient for the needs of the article submitted for review. The authors have demonstrated that the application of foliar fertilization with NPs of Se can be an alternative to improve the production of higher quality forage crops.

I find this manuscript valuable and interesting because of the cognitive values it brings, nevertheless I have some comments on it.

Keywords should be different from the words in the title.

Materials and methods. 4.10. Measurement of Selenium Content in Grass. Was the certified reference material analyzed with the samples in this study? This is not included in this chapter and Se recovery rate is not reported.

If the article does not contain a separate Discussion chapter, it should be clearly indicated that the results are discussed in the Results section anyway, and thus the chapter should be called Results and Discussion.

Line30-33: Twice is the same sentence.

Line 112: It's ND it should be NDF.

Line 186: In parentheses instead of tannin content are selenium content. Compare Table 3.

Line 232: Do the results in parentheses really apply to ABTS?

Line 250-251: There should be full Latin species names.

Table 3. Total Phenols - instead of mg/100g DM please write mg GAE/100g DM; Total flavonoids - instead of mg/100g DM please write mg QE/100g DM; Total tannins - instead of mg/100g DM please write mg EC/100g DM.

Author Response

Thank you for your comments; 

We have already corrected all your suggestions,

Regards 

Gabriela 

Author Response

Thank you very much for your observations
I appreciate your comments; we have already made the respective corrections.

Sincerely
Gabriela

Reviewer 3 Report

The aim of the study was to evaluate the effect of foliar application of selenium nanoparticles in a forage on the biomass, chemical composition and antioxidant content.

There have been many studies on the effects of selenium nanoparticles on plant nutrition and physiology, but there is no information on the effects on the toxicity and other aspects of selenium nanoparticles use in agriculture. The topic of the work is interesting from a practical point of view.

The analytical methods used in the manuscript are fully described, as is the experimental design. The use of mainly spectrophotometric methods for determining the group of compounds such as polyphenols, tannins or flavonoids lowers the value of the obtained results. It would be more interesting to observe changes in the content of individual compounds, which would only be possible with the use of chromatographic methods.

The text of the manuscript is not adapted to the requirements of the journal and needs to be completely changed. Chapter 3, which is a discussion of the results obtained, is missing. Partial discussion of the results is presented in the description of the obtained results, but it is insufficient. I recommend adding a separate chapter, which will present a discussion of the results, also based on previously published works.

The applied statistical method of comparing the results also requires explanation. Why was the two-factor analysis method (Table 1) used to assess the effect of foliar application of Se NPs on the morphological characteristics of grass plants, and why was the one-factor analysis of variance used to assess Nutritional characterization of the grass (Festuca arundinacea) bio-fortified via foliar with different concentrations of NPs of Se (Table 2)? Why were the factors reduced if the experience was the same? This requires some explanation.

Detail comments:

L307 - hours Subsequently - no dot

L190 - the grass. (Festuca arundinacea). - too many dots

Author Response

Thank you very much for your comments, we have already made the corrections
Our manuscript has improved a lot.

Best regards,
Gabriella

Reviewer 4 Report

Major comments

  1. Traditional bioforitification of forage crops with selenium is utilization of soil (Finnish experiment) or foliar (Slovenia) application of sodium selenate. There is no explanation why Se nano-particles have been chosen and no comparison with any other chemical form of selenium. How expensive is the utilization of nano-Se compared to sodium selenate? What are the benefits of nano-Se? I have not found any references concerning fundamental research in Finland on biofotification of forage crops (at least read and cite: Alfthan G, Eurola M, Ekholm P, Venäläinen ER, Root T, Korkalainen K, Hartikainen H, Salminene P, Hietaniemi V, Aspila P, Aro A (2015) Effects of nationwide addition of selenium to fertilizers on foods, and animal and human health in Finland: from deficiency to optimal selenium status of the population. J Trace Elem Med Biol 31:142–147. https://doi.org/10. 1016/j.jtemb.2014.04.009)
  2. The authors indicated that they used three different crops (see lines 16 and 284) What does in mean: ‘Three times of harvest of the grass were established’ Was it a repetition or samples were obtained continuously at a certain intervals? Then what intervals? If it were repetitions, then the authors should use means in all Tables and appropriate concentration range and cv, %.  It is impossible to understand from the Discussion and Material and Methods sections whether these were three repetitions or three consistent harvesting of forage grass.

3.Material and Methods section should include more detailed data on the determination of fiber (CF, NDF and ADF) so that the reader will be able to understand the peculiarities of the analysis

  1. No data are available as to the importance and significance of ether extraction- why the authors have made these analyses?
  2. In all scientific investigations it is highly important to show the degree of different parameters changes but not use only absolute values
  3. Taking into account the concentrations of Se in plants it seems interesting to calculate the approximate value of Se consumption with domestic animals

Minor comments

  • Revise all the text and use one style of units: mg m-2 instead of ‘mg / m2’ (see line 17), etc (lines 28 and further…
  • If you speak about ether extract (line 19) then please indicate its significance
  • line 27 ppm / Kg DM     - impossible, use ppm only without Kg
  • line 38- change ‘phenols’ to ‘phenolics’’
  • line 55 ‘alfalfa that lack selenium’- change to ‘trace levels’
  • line 70 ‘NPs have different physicochemical properties compared to bulk materials’- too common words. It is necessary to indicate NP beneficial effects
  • line 76 change ‘Results’ to ‘Results and discussion’
  • line 132 ‘The decrease in CF can be attributed to the fact that Se-treated plants have been shown to increase the concentration of soluble sugars’- add a citation
  • Line 139 ‘reportby' change to ‘report by’
  • Table 3- decipher I,II,III in Table notes
  • Line 195 ‘A study [40] reported in a study where’..- check style
  • Use Se abbreviation everywhere instead of ‘selenium’
  • Line 218 ‘The low consumption of thyroid hormone metabolism and the immune response is altered’- think that it should be ‘the low consumption of selenium…’- check grammar
  • Line 223 ‘They found an increase in Se content due to foliar enrichment of Se’- it is obvious, delete
  • Line 224 style: ‘Chen et al [43] enriched rice with foliar via sodium selenite and sodium selenate. The effects of the fertilization were higher when selenite (0.471 ± 0.134 µg g−1 ) and selenate (0.640 µg g−1 ) were applied compared to the control (0.071 ± 0.002 µg g−1 ).’ You are speaking about Se concentrations- the style is awful. But in general, why do you use this citation?- there are no data on nano Se
  • Line 230 ‘The fortified foliar treatment with a concentration of 4.5 mg / m2 NPs presented the highest antioxidant capacity 231 (284.26 ± 1.8, 278.35 ± 2.2 and 289.96 ± 2.6 mg / EEE) with significant differences against the other treatments (p<0.05).’- it is highly desirable to indicate the values of antioxidant activity increase
  • Materials and methods: indicate the area of the experimental site
  • Line 371 ‘ how do you manage to heat the reaction mixture with HN03 at 175 oC during 1 hour, when the boiling temperature of HN03 is only 83oC???
  • Data availability statement- revise
  • Reference list: use (i) journals abbreviations, (ii) bold letters for appropriate year of publication and (iii) italics for Latin names of plants (iiii) revise the reference list according the author guidelines (for instance ref.1 ‘Chen L, Auh CK, Dowling P, Bell J, C..’ change to  ‘Chen, L.; Auh, C.K.; Dowling, P.; Bell, J.;..’, etc

Author Response

(The authors gave the same response as above.)

Round 2

Reviewer 3 Report

My comments have been incorporated in the revised version of the manuscript